# Sparse Fourier Backpropagation in Cryo-EM Reconstruction

**Dari Kimanius**          **Kiarash Jamali**          **Sjors H.W. Scheres**

MRC Laboratory of Molecular Biology
{dari, kjamali, scheres}@mrc-lmb.cam.ac.uk

## Abstract

Electron cryo-microscopy (cryo-EM) is a powerful method for investigating the structures of protein molecules, with important implications for understanding the molecular processes of life and drug development. In this technique, many noisy, two-dimensional projection images of protein molecules in unknown poses are combined into one or more three-dimensional reconstructions. The presence of multiple structural states in the data represents a major bottleneck in existing processing pipelines, often requiring expert user supervision. Variational auto-encoders (VAEs) have recently been proposed as an attractive means for learning the data manifold of data sets with a large number of different states. These methods are based on a coordinate-based approach, similar to Neural Radiance Fields (NeRF), to make volumetric reconstructions from 2D image data in Fourier-space. Although NeRF is a powerful method for real-space reconstruction, many of the benefits of the method do not transfer to Fourier-space, e.g. inductive bias for spatial locality. We present an approach where the VAE reconstruction is expressed on a volumetric grid, and demonstrate how this model can be trained efficiently through a novel backpropagation method that exploits the sparsity of the projection operation in Fourier-space. We achieve improved results on a simulated data set and at least equivalent results on an experimental data set when compared to the coordinate-based approach, while also substantially lowering computational cost. Our approach is computationally more efficient, especially in inference, enabling interactive analysis of the latent space by the user.

## 1 Introduction

Single-particle electron cryo-microscopy (single-particle cryo-EM) has become the fastest growing method for structure determination of biological macromolecules in recent years, and is becoming an increasingly important cornerstone of molecular biology research and drug discovery [1, 2, 3, 4]. Briefly, a thin layer of an aqueous solution containing multiple copies of the target molecule in random orientations is flash frozen to cryogenic temperatures and imaged in a transmission electron microscope. Subsequently, two-dimensional projection images of the Coulomb scattering potential of the molecules in the sample can be acquired into a micrograph. Micrographs typically contain information to sub-nanometer details of up to millions of copies of the target molecules that are called particles.

To limit damage to the radiation-sensitive particles, micrographs are acquired under low-dose conditions, which leads to low signal-to-noise ratios (of typically less than 1/10) which requires averaging over many images [5]. If a sufficient number of approximately identical particles are captured in distinct projection orientations (or poses), computational methods can be employed to reconstruct a three-dimensional volume of the scattering potential of the particle. However, the poses of the individual particles are unknown and many particles adopt multiple conformational states, or assemblies of multiple molecules may vary in their composition. The unknown poses and the potential

36th Conference on Neural Information Processing Systems (NeurIPS 2022).

assignments to multiple different structural states renders the problem ill-posed. Combined with the low signal-to-noise ratio in the images, this makes single-particle cryo-EM reconstruction a challenging task.

The reconstruction process starts with the extraction of a large number of image patches (on the order of $10^4 - 10^7$) from the micrographs, each containing a single projection image of the particle, captured in an unknown pose. In order to perform a three-dimensional reconstruction, the poses of all particles need to be determined, and particles need to be separated into structurally homogeneous subsets. Typically, this task is performed in two stages. First, in a classification stage, structurally homogeneous subsets are identified and undesirable particles are removed. Second, in a refinement stage, the selected subsets of particles are reconstructed to obtain a high-resolution structure.

Although generally desirable, automation and high throughput are essential in several areas of application, e.g. fragment-based drug discovery [4] or screening for disease-relevant structures [6]. Fast computational processing pipelines, based on hardware acceleration [7, 8, 9], have proven critical to keep up with the high data rates provided by automated image acquisition software [10]. However, classification remains a bottleneck in existing pipelines, often requiring expert user input and repeated trials. Therefore, unsupervised algorithms that can replace existing classification steps will be essential in further improving single-particle cryo-EM. Moreover, because conformational changes are often central to the biological function of proteins, more powerful methods to deal with structural heterogeneity have the potential to enable biological discoveries which are currently out of reach of the existing methods.

## 2 Background and Definitions

### 2.1 Cryo-EM Reconstruction

Let $\{y_i \in \mathbb{C}^{M^2} : 1 \leq i \leq N\}$ be a set of images of (Fourier transformed) individual particles, where $M$ is the image size and $N$ is the number of particles. Each $y_i$ is modelled as a projection of a three-dimensional signal. The projection operation $\mathcal{H}(\phi_i)$ is described by the pose of the particle, $\phi_i = (R_i, t_i) \in \mathrm{SE}(3)$, where $R_i \in \mathbb{R}^{3 \times 3}$ is a rotation or reflection matrix and $t_i \in \mathbb{R}^3$ is a translation. Through the Fourier-slice theorem, the projection operation is realized by taking a central slice from the three-dimensional Fourier transform of the signal, $V \in \mathbb{C}^{M^3}$. The projected image is then convolved with the point spread function of the optical system, which is described in the Fourier domain by a multiplication with the *contrast transfer function* (CTF), $\mathcal{C}(\kappa_i)$, with known parameters $\kappa_i$. Finally, experimental noise is modelled as additive and independent Gaussian in the Fourier domain, with a scale that is frequency-dependent, $\varepsilon \sim \mathcal{N}(0, \sigma_y)$. Thereby, the forward model becomes:

$$y_i = \mathcal{C}(\kappa_i) \circ \mathcal{H}(\phi_i) \circ V + \varepsilon \tag{1}$$

If the data set is structurally homogeneous, all particles are projections of a common $V$, and the objective is to estimate this $V$ from all $y_i$, with unknown $\phi_i$ and known $\kappa_i$. If the data set is structurally heterogeneous, there is no longer a single $V$ to solve for. Rather, the solution is some space $\mathcal{V}$ and each $y_i$ corresponds to a $V \in \mathcal{V}$. In either case, we can represent the task at hand as the solution to a maximum-a-posteriori estimate, where we marginalize over the nuisance variable $\phi$:

$$\hat{\mathcal{V}} = \underset{\mathcal{V}}{\mathrm{argmax}} \left( \sum_{i=1}^{N} \log \int_{\mathcal{V}} \mathbb{E}_{\phi \sim \mathrm{P}(\phi|y_i, V)} \Big[ \mathrm{P}(y_i|V, \phi) \Big] dV + \mathcal{P}(\mathcal{V}) \right) \tag{2}$$

In the above, $\mathcal{P}(\mathcal{V})$ represents a log prior over the solution space. A conventional approach to solving this problem is clustering the space $\mathcal{V}$ to a discrete set of $K$ structural states, which means that the integral in (2) becomes a sum over a fixed $K$. Popular software packages for single-particle cryo-EM, like *RELION* [11] and *cryoSPARC* [9], employ iterative algorithms, including expectation-maximization and gradient decent, to find a solution for the problem with $K$ distinct structural states. For computational reasons $K$ is typically kept below 10. The expectation over $\phi$ is also computationally expensive, which is why approximate approaches, including branch-and-bound methods [9], or variations thereof [12], are regularly used.

## 2.2 Cryo-EM Generative Modelling

By parametrizing $\mathcal{V}$ differently, the cryo-EM reconstruction task can also be described with generative modelling. Using the autoencoder framework, let $f_\theta$ be the encoder parameterized by $\theta$, and $g_\omega$ be the decoder parameterized by $\omega$. Each particle image $y_i$ can be embedded in some latent encoding $z_i \leftarrow f_\theta(y_i)$. Then, using the decoder, we reconstruct the original $y_i$ by minimizing a loss function $\mathcal{L}$ that will contain some priors and a log-likelihood term $\left\| y_i - \mathcal{C}\left(\kappa_i\right) \circ \mathcal{H}\left(\phi_i\right) \circ g_\omega(z) \right\|_2$. To reduce the computational cost of generative models, the expectation in (2) can be replaced with the highest probability pose for each particle, $\hat{\phi}_i$, as determined using a homogeneous reconstruction approach.

The generative modelling approach enables replacing the fixed number of discrete classes, $K$, with a model that can account for a larger variability in $\mathcal{V}$. This is especially relevant when dealing with structural heterogeneity that cannot be easily separated in distinct states, like continuous motions. Protein complexes often populate states that represent continuous transitions as part of their functional cycle. The continuous distribution of the learnt latent representation, expressed by $g_\omega$, provides a natural representation of the corresponding protein conformational landscape. Assuming that the experimental procedure does not perturb the distribution of structural states in the sample, the latent representation could provide insights into the dynamics and perhaps the biological function of the protein.

Moreover, although not strictly equivalent, the expansion of model capacity provided by the generative modelling approach can be approximated in the discrete setting by increasing $K$ to a large number. It has been shown that robustness of canonical non-parametric classifiers are critically dependent on a large $K$ [13]. Therefore, we expect generative models with larger model capacity to improve robustness over discrete heterogeneous reconstruction methods with low $K$.

## 3 Related work

The generative modelling approach described in the previous section can be classified based on the space where the reconstruction takes place; either one decodes the volume in Fourier-space or in real-space. Both methods have their own benefits, as described below.

### 3.1 Fourier-space reconstruction

*cryoDRGN* [14] is currently the most popular approach to Fourier-space reconstruction using the generative modelling framework. In cryoDRGN, a variational autoencoder (VAE) is used to encode particle image heterogeneity into a Gaussian parameterized latent space. In particular, cryoDRGN encodes each particle image with $\mu_i, \sigma_i \leftarrow f_\theta(y_i)$. To obtain the latent space encoding, they sample $z_i \sim \mathcal{N}(\mu_i, \sigma_i)$. Next, they independently reconstruct all Fourier components that correspond to the specific pose, $\phi_i$, of the particle image using a coordinate-based decoding function $g_\omega(z_i, \boldsymbol{\xi})$, where $\boldsymbol{\xi}$ describe the 3D coordinates. For this approach to work, the hidden variables, including the projection angle, are determined beforehand through global angular searches (see section 2.1). Similar to neural radiance fields (NeRF) [15], the decoder is typically a multilayer perceptron (MLP) that receives a positional embedding of the coordinates of each Fourier component of the image and its latent vector.

Another approach to Fourier-space reconstruction using the generative modelling framework is *3D variational analysis* (3DVA) [16]. This method does not use deep learning or gradient-based methods for learning its parameters, but starts from a reconstruction and formulates the generative process as a probabilistic principal components analysis to obtain a set of orthogonal basis $V^{(k)}$ of 3D Fourier volumes representing the principal components that best fit the heterogeneity. This method optimizes the log-likelihood with an expectation-maximization approach.

Due to memory restrictions, cryoDRGN cannot reconstruct full volumes in a single batch, since the reconstruction of each coordinate in the volume requires a pass through the decoder. This prohibits backpropagation with respect to the full volume because the intermediate activations for $\mathcal{O}(M^3)$ coordinates have to be stored. This quickly becomes intractable. Although, this issue can be ameliorated with gradient checkpointing [17], it comes at the cost of extra training time. This hinders the application of real-space priors, e.g. solvent masks, mass preservation, and data-driven priors [18].

3DVA does not have this issue since full volume reconstructions are more computationally efficient than they are with cryoDRGN, as volumes are weighted averages of the principal components and the original reconstruction. However, in contrast to 3DVA, a VAE can be trained in a fully differentiable manner, which confers unparalleled flexibility to the method and allows easy modification of the loss function. For instance, one can easily add an auxiliary loss that imposes additional constraints to either the latent space or the output. It's noteworthy that gradient descent optimization has shown superior convergence properties over that of expectation-maximization for *de novo* reconstruction [9]. Furthermore, since a VAE is trained to do amortized variational inference, the encoder can generalize to previously unseen data without the need for re-training. The approach presented in this paper is based on a VAE framework and hence enables both easy modification of the loss function and amortized variational inference with an encoder. On the other hand, we modify the decoder to gain some of the computational benefits of the 3DVA approach, which combined with the VAE framework enables fast and efficient full volume inference and differentiation.

## 3.2 Real-space reconstruction

*3DFlex* [19] is a method implemented in the cryoSPARC software package that is based on real-space reconstruction. In this approach, a consensus real-space reconstruction is learnt and then deformed by a flow field that is parameterized by an MLP, given a latent representation for each image. Instead of viewing the problem as generating independent volumes, it deforms one consensus volume such that it maximizes the log-likelihood of the individual particle images.

Another real-space based method, *e2gmm* [20], also learns flow fields, but these are applied to a Gaussian mixture model instead of a volumetric representation. In this method, the encoding is done similar to cryoDRGN, where particle images are encoded in a Gaussian latent space.

Both real-space approaches enforce a mass-preserving inductive bias, which can improve reconstruction quality. Although not yet explored, both approaches would in principle enable a computationally efficient application of other real-space priors, like solvent masking or data-driven priors.

A disadvantage of the real-space methods is that the application of the CTF requires two Fourier transforms (forward and backward) to evaluate the log-likelihood loss. To avoid expensive 3D Fourier transforms, the implementation of both of the above mentioned methods first project along the orientation of each particle image and then apply a 2D Fourier transform. However, the projection operation is computationally expensive in real-space, especially when dealing with volumetric representations, because the entire volume has to be accessed.

This contrasts with Fourier-space methods, where the CTF can be applied by multiplication and the projection operation can be performed by accessing a small fraction of the Fourier components in the volume, due to the Fourier-slice theorem. To exploit the latter, in what follows we present an approach where the decoder is not a function of the 3D coordinates, but rather generates the entire volume in Fourier-space in a single pass, and present a novel approach for training such a model in a computationally efficient manner.

## 4 Method

Here, we present an approach for learning a volumetric Fourier reconstruction given a latent representation in a VAE framework. Our decoder learns to generate an entire volume for each latent vector. This volume consists of the model Fourier coefficients on a regular cubic grid with fixed coordinates. To enable efficient training, we employ a sparse backpropagation method that limits evaluation and gradient calculations, in both the forward and backward pass, to the subset of model parameters that are affected by each projection image. This is implemented in *PyTorch* [21] with custom *CUDA* kernels. We have not preformed any runtime optimizations on these kernels. The code used in this paper is available at github.com/dkimanius/sbackprop, together with the instructions for compiling and running it.

### 4.1 Motivation

As previously noted, cryoDRGN uses a coordinate-based decoder, similar to NeRF, to parameterize the reconstructed volumes. However, unlike the real-space case to which NeRF is applied, cryoDRGN's

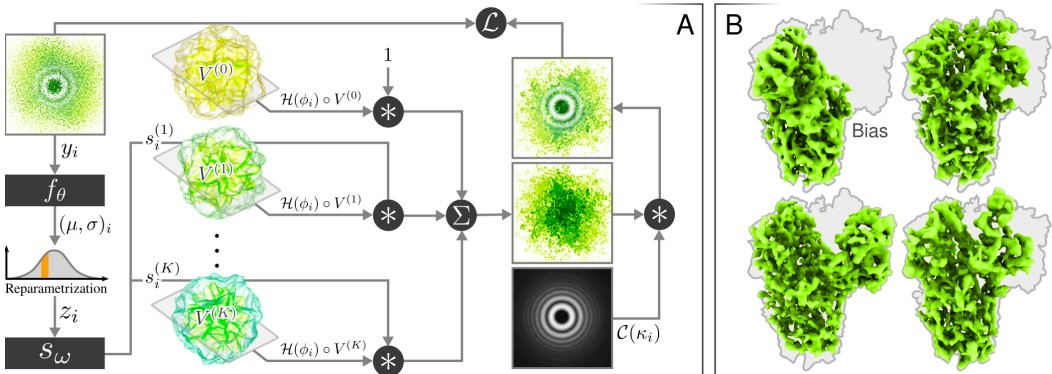

Figure 1: **(A)** shows the sparse backpropagation method. The projection image $y_i$ is input into the encoder $f_\theta$ which results in a set of scalars $s_i^{(k)}$ that weigh the learnable and interpretable structural basis set $V^{(k)}$. The weighted sum of the projection $\mathcal{H}(\phi_i)$ is then CTF corrupted with $\mathcal{C}(\kappa_i)$. Finally, the input $y_i$ and the output are compared in the loss function, $\mathcal{L}$. **(B)** shows the bias parameter and two structural bases of the model in real-space for the reconstruction in section 5.1

outputs are defined in Fourier-space, where the spatial smoothness assumption is less valid. In particular, this parameterization suffers from the spectral bias of neural networks [22, 23]. That is, the inability of neural networks to capture local, high-frequency changes. This makes Fourier-space representations more challenging to learn with a coordinate-based approach. This issue can be ameliorated by incorporating a high-frequency positional embedding [24], but an alternative approach that avoids positional embedding altogether might be better suited for this particular problem. A potentially more natural approach to learning discrete Fourier representations is on a regular cubic grid, a so called voxel representation. Similar approaches in real-space have been shown to improve model capacity and training efficiency [25]. In particular, we suggest to learn a set of volumes $\{V^{(k)} \in \mathbb{C}^{M^3}\}$ that are combined linearly into the final output.

While potentially better behaved, this method would ultimately result in the need to store $B \times K$ volumes for the backward pass during training, where $B$ is the batch size. This is computationally wasteful and has a large memory footprint, which is why we propose a novel training method. The following sections outline how we exploit the sparse property of the cryo-EM forward model in Fourier-space to more efficiently evaluate and train volumes $V^{(k)}$ with backpropagation.

## 4.2 Structural Basis

We define the decoder as

$$g_\omega(z) = \sum_{k=1}^{K} V^{(k)} s_\omega^{(k)}(z) + V^{(0)}. \tag{3}$$

Here, $V^{(k)} \in \mathbb{C}^{M^3}$ are trainable volume-sized arrays that we refer to as *structural basis*; $s_\omega^{(k)} \in \mathbb{R}$ are the latent-dependent weighting factors; and $V^{(0)}$ is a trainable bias term. $K$ is strongly correlated with the expressive capacity of the model. Together with the volume size, $K$ has a major impact on the total memory footprint. The volume size is determined by the data image size, i.e. by the image physical pixel size and the maximum resolution at which the model needs to be evaluated. For all experiments in this work we use $K = 16$; $s_\omega$ is a small three layer MLP with residual connections [26].

This approach has similarities to 3DVA, in that volumes are linearly combined to generate the reconstruction results. However, in our approach the structural bases are learnt together with the representation in an end-to-end differentiable setup, including an encoder parameterized by a neural network. The VAE is trained to perform amortized variational inference and we use gradient based optimization, which makes the computational burden comparable to cryoDRGN rather than 3DVA (see section 3.2).

## 4.3 Training

For the loss we use the log-posterior expression used in the RELION-4.0 implementation [27], which is

$$(1-\lambda)\frac{\mathcal{C}^2\left(\kappa_i\right)}{\sigma_y^2}\left\|y_i - \mathcal{C}\left(\kappa_i\right) \circ \mathcal{H}\left(\phi_i\right) \circ g_\omega(z_i)\right\|_2^2 + \lambda \mathcal{P}(g_\omega(z_i)). \tag{4}$$

Here, $\lambda$ is a frequency-dependent regularization parameter; $\mathcal{P}$ is a suitable log-prior; and $\sigma_y$ is the frequency-dependent noise amplitude of the data. In this work we set $\mathcal{P} = 0$.

During training, the Fourier slice belonging to the known projection plane of each data sample (particle) is extracted from the cubic grids of the structural bases, using tri-linear interpolation, similar to [28]. The extracted slice is then real-space shifted through phase-shifts in Fourier-space, and modulated with the CTF, in order to calculate its log-likelihood loss. Although the projection angle, shift, and CTF parameters are given, it is possible to jointly optimize them with the model parameters, since all the operations involving them are differentiable. However, in this paper we will keep them fixed.

Since the sparse backpropagation method generates gradients for a subset of the model parameters, it is natural to use the sparse version of the *Adam* optimizer [29], where the gradient moments are only updated for the affected parameters. This requires that the gradients are stored in a sparse representation, which is less memory efficient for large batch sizes. However, since each batch contains particles from different orientations, when the batch size is sufficiently large, enough of the volume would receive gradients, which enables the use of dense gradients. This makes the memory footprint of the sparse projection operation invariant to batch size.

Because of the low signal-to-noise ratio in cryo-EM data, we can approximate the CTF corrected $\sigma_y$ as the averaged spectral amplitude of the data divided by the average CTF amplitude, here denoted as $\bar{\sigma}_y$. Let $\mathcal{B} = \{\boldsymbol{\xi} : \xi < |\boldsymbol{\xi}| < \xi + \tau\}$, where $\tau$ depends on the discretization used for the discrete Fourier transform. Then, for a frequency $\xi$ we get

$$\bar{\sigma}_y^2(\xi) = \frac{\sum_{i=1}^N \int_{\mathcal{B}} \left\|y_i(\boldsymbol{\xi})\right\|_2^2 d\boldsymbol{\xi}}{\sum_{i=1}^N \int_{\mathcal{B}} \mathcal{C}^2\left(\boldsymbol{\xi}, \kappa_i\right) d\boldsymbol{\xi} + \epsilon}. \tag{5}$$

In the above, the integral is evaluated over the frequency shell with radius between $\xi$ and $\xi + \tau$ and the sum is over all the images in the data set. The value of $\epsilon$ is set to a small positive number for numerical stability.

Although the real-space particle images are standardized to zero-mean and standard deviation of one, the numerical values of their Fourier transforms typically have values spanning several orders of magnitude. To improve convergence of the model parameters, we perform a spectral normalization of the Fourier transforms of the data samples, where the normalization factor is a function of Fourier frequency. In this work, we use $\bar{\sigma}_y$ as the spectral normalization factor, which makes the effective $\sigma_y$ in (4) approximately equal to one. To reverse the effect this has on the spectral weighting of the MSE-loss, we set $1 - \lambda$ proportional to $\bar{\sigma}^2$.

We note that it is the sparse access of the projection operation $\mathcal{H}(\cdot)$ that enables an efficient evaluation and training of $g_\omega$. For the sparse backpropagation method, we define a single fused operation that evaluates both $g_\omega$ and $\mathcal{H}(\cdot)$ in a single function call. In the forward pass, this operation disregards parameters in $g_\omega$ that are omitted by $\mathcal{H}(\cdot)$. Similarly, in the backward pass, gradients belonging to omitted parameters are not calculated.

Additionally, to minimize the memory footprint, we only store Fourier components in $V^{(k)}$ in a sphere with a radius corresponding to the Nyquist frequency, as defined by the user. This requires explicit coordinate arrays, which negatively impacts performance, but reduces the memory footprint by almost a factor of two, compared to storing Fourier components of the entire cube.

We evaluate our method on one simulated data set and an experimental cryo-EM data set, which was downloaded from the publicly available Electron Microscopy Public Image Archive (EMPIAR) [30], with a CC0 licence [31]. All calculations were performed on an NVIDIA RTX 3090.

# 5 Results

## 5.1 Simulated data set

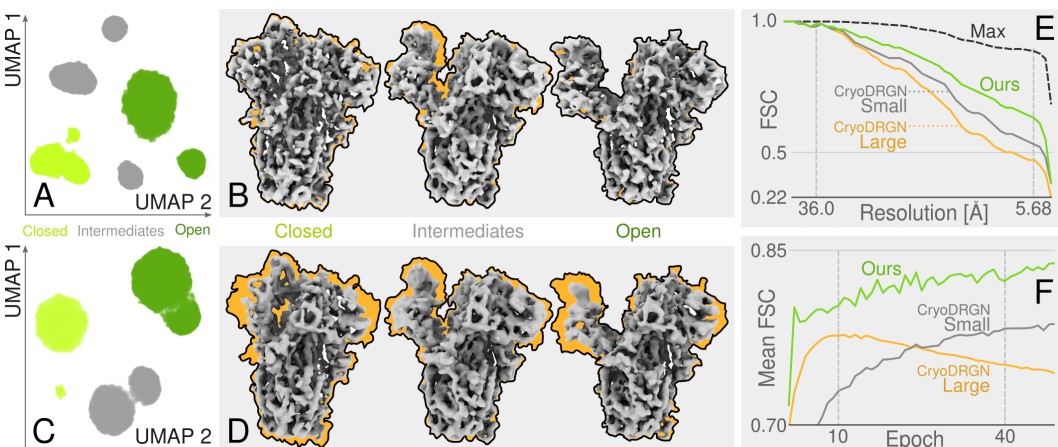

Figure 2: Reconstruction results of the simulated data set of SARS-CoV-2 spike protein after 50 epochs of training. **(A)** shows the learnt latent representation of our method versus **(C)** cryoDRGN-small. **(B)** shows the side view of example reconstructions for our method and **(D)** shows the same for cryoDRGN-small. The outline of each ground truth structure is shown with a black contour and orange fill to highlight discrepancies between the reconstructions and the ground truth. **(E)** shows the average FSC to each ground truth for our method (green) versus the cryoDRGN-small (grey) and cryoDRGN-large (orange). The dashed line shows the maximum possible FSC. **(F)** shows the average FSC per epoch. The average of the maximum FSC shown in (E) is 0.95.

The simulated data set consists of projection images of 3D volumes generated from frames of a molecular dynamics simulation of the SARS-CoV-2 spike protein (DESRES-ANTON-10897850) [32], published under a CC BY 4.0 license [33]. Ten different conformational states were extracted out of the simulation and ten thousand projection images with random poses were generated for each state, amounting to 100,000 images in total. Three frames were selected of the so-called open state; three frames of the closed states [34]; and four frames of the intermediate states between the open and the closed states. The image size is $M = 96$, with a physical pixel of 2.7 Å. The simulated images were then distorted with additive Gaussian noise and CTF. The average signal to noise ratio is 1/10 and the defocus range is between 0.2 and 2 µm. The pose for each particle image was then estimated by running a RELION consensus refinement, and kept fixed in the following steps. This data set, along with the PDB files used to create it and the refinement results, can be found at zenodo.org/record/7182156 (DOI:10.5281/zenodo.7182156).

We trained our VAE with 8 latent dimensions for 50 epochs. After the end of training, we applied uniform manifold approximation and projection (UMAP) [35] to the distribution of each particle image's latent vectors. Evaluating our decoder for points belonging to each cluster revealed that several distinct conformations where separated for each of the three states (see figure 2A). In total seven separated clusters, were observed, amongst which, four belong to single conformations and three are made up of mixtures of at most two conformations. CryoDRGN only fully separated one distinct conformation, belonging to the closed state. The rest remained in clusters mixed with other conformations (see figure 2C).

Running cryoDRGN on this data set with the default network architecture (cryoDRGN-small) and a larger one (cryoDRGN-large) results in worse reconstruction performance compared to our method, as can be seen in figure 2. Examples of reconstructions from the final epoch for our method and cryoDRGN-small are shown in figure 2 panel B and D, respectivly. Since this is simulated data, we can compare the results to the ground truth. Comparison to the outline of the ground truth, reveals that the cryoDRGN results have more missing structural features and overall lack the same level of details seen in the results of our method, particularly in the flexible region. Furthermore, we can calculate the Fourier shell correlation (FSC) between the generated and the ground truth structures. The FSC is calculated here as the normalized complex correlation between Fourier components of the solvent

masked reconstruction and the known ground truth for each frequency band [36], averaged over 30 samples (three out of each conformation). The results for this analysis are visualized in figure 2E. At the Nyqvist frequency (5.4 Å resolution) the FSC drops due to aliasing [37]. This is also visible for the maximum FSC, which is calculated by reconstructing with the known conformational and pose assignments. Figure 2F shows the average FSC to ground truth for each epoch. Neither of the methods converged in 50 epochs. However, starting around epoch 10, cryoDRGN-large diverges from ground truth, which is seen as the average FSC dropping, due to overfitting to noise. Additionally, increasing batch size gave worse results for cryoDRGN, as can be seen in the in supplementary table 1 and supplementary figure 5.

It took cryoDRGN-small 213 minutes to train for 50 epochs, while our method takes 40.3 minutes. Evaluating the decoder takes 5 ms on average for each 3D structure, which is on the same order of magnitude as the time for the inverse Fourier transform of 1 ms. This can be compared to 250 ms for the cryoDRGN-small architecture.

### 5.2 Ribosome data set

We also applied our method to an experimental cryo-EM data set of 80S ribosome particles from the *Plasmodium falciparum* parasite [38]. This data set has been used repeatedly as a test case for new algorithms [39, 14, 40, 41]. We used 108,732 particle images, with $M = 160$ and a pixel size of 3.0Å. We used 10 latent dimensions for this data set. To evaluate the ability to handle defective particle images, we also included 78,442 images that where classified by RELION as not containing ribosomes, or containing partially (dis-)assembled ribosomes, along with images classified as intact ribosomes. This brings $N$ to 187,174.

After a few epochs, the defective and empty particles were separated out as the top principal component and the separation is clearly preserved in a UMAP of the latent space after 50 epochs (figure 3C). Additionally, we notice separation of different classes of particles in each of the two major classes. In the cluster of the intact ribosomes we observed several subtle continuous conformational transitions, including the "head swivel" [42] (not shown in figure) and the rotation of the small subunit, which is part of the "ratcheting" cycles [43] (figure 3B). Although the figure shows the two endpoints of the rotation, a continuous transition between the two states can be observed along the marked path in the latent representation. In the other cluster, containing the defective particles, we observed varying classes of particles. Images of partially assembled ribosomes are separated from images lacking any protein features (figure 3D).

Next, we trained cryoDRGN-large on this data set and achieved similar results to our method, as can be seen in figure 3E-H. Unlike in 5.1, we do not have access to the ground truth volumes in order to compute an FSC. However, qualitatively, we are able to generate similar clustering and reconstruction performance compared to cryoDRGN, at lower computational cost. It should also be noted that training cryoDRGN-small on this data set gives noticeably worse results, as shown in the appendix. Training took 210 minutes using our method, compared to 1638 minutes for cryoDRGN. Moreover, evaluating the decoder takes 10 ms on average for each 3D structure, compared to 480 ms for cryoDRGN.

We kept the batch size of cryoDRGN to the recommended value and what gave the best results for the simulated data set (see section 5.1).

## 6 Discussion

The potential improvement in robustness make VAE approaches a candidate for replacing the discrete classification methods that are commonly used in cryo-EM single-particle reconstruction pipelines. We identified the increased computational cost of deep learning based methods as an issue preventing this shift, and present an alternative method with potential to solve this issue.

Our results on simulated data illustrate that our method yields improved results compared to cryoDRGN with less than one-fifth the computational cost. We show that our method yields reconstructions with higher correlation to the known ground truth. For an experimental test case, our method converges onto a solution that is at least as useful as the one provided by cryoDRGN, with one-eighth of the computational cost and a smaller memory requirement.

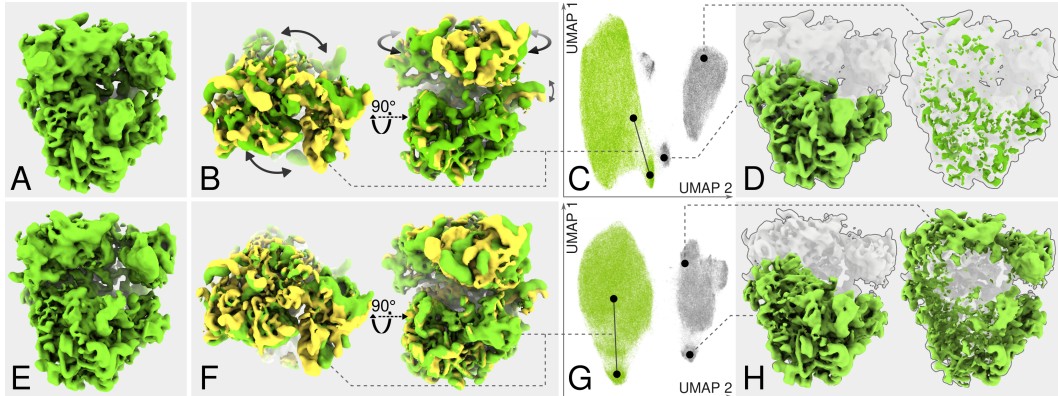

Figure 3: Reconstruction results of the Plasmodium falciparum 80S ribosome data set after 50 epochs of training using our approach **(A-D)** or cryoDRGN-large **(E-H)**. **(A, E)** Reconstruction of the ribosome, showing the small and large subunits. **(B, F)** Two structures (green and yellow) showing the two rotational states. **(C, G)** The learnt latent space, dimensionality-reduced with UMAP. Clusters containing particle images of intact ribosomes marked in green and empty images or partially assembled ribosomes in grey. **(D, H)** Two reconstruction examples from the grey cluster, with a ribosome lacking the small subunit on the left and an empty particle containing mostly noise or low contrast features on the right.

From the user's perspective, our approach also offers a practical advantage in allowing a faster interactive analysis of latent space. Calculation of a volume for a given point in latent space takes on the order of half a second in cryoDRGN. Our decoder reconstructs entire volumes in one pass, which takes only a few milliseconds. This allows for a more convenient interactive analysis of latent space, where the user inspects multiple volumes on-the-fly, along interactively selected points of interest in latent space. Intuitive graphical tools enable inexperienced users to explore the heterogeneity in their data, and easily identify particle subsets to be used for subsequent refinements. This holds the potential to make cryo-EM accessible to a broader range of researchers.

This approach can also be combined with global angular searches because of the reduced overall computational cost and the fast full-volume generation (see section 3.1). This could enable reconstruction of data sets where the initial angular assignment based on homogeneous reconstruction fails. Additionally, this could accelerate the replacement of the discrete 3D classification where global angular searches are integrated in all popular implementations.

Because of the low signal-to-noise ratios in the experimental images, cryo-EM reconstruction is prone to overfitting, and explicit regularization of the output maps has played a major role in the most popular software packages [11, 9]. Operations like (back-)projection and the application of the CTF are performed conveniently in Fourier-space. Most software packages thus perform reconstruction in Fourier-space, where they also regularize the power of the Fourier components of the reconstruction. Our approach, which also performs reconstruction in Fourier-space, is well suited to apply similar regularization in Fourier-space, although we did not explore this here. Moreover, because reconstructed maps are generated at relatively low computational cost, one could also apply real-space priors, like solvent masks or data driven priors [18]. Because these priors are differentiable, one could backpropagate through them to improve this method's structural bases and network parameters. In this way, one could directly use the data-driven prior to improve the reconstruction in a computationally efficient manner. Improved algorithmic robustness through the use of priors on the reconstructed set of volumes has the ability to increase the radius of convergence of the reconstruction algorithm and allow more challenging data sets to be analyzed.

The structural basis, and the linear manner in which they are combined, also allow for better interpretability of what the model is learning. In addition to facilitating troubleshooting, the structural basis can potentially be utilized for two purposes. Firstly, they could be used to impose 3D priors on the reconstruction in a more direct manner than backpropagation. Real-space priors that are invariant to structural heterogeneity, e.g. solvent masks or structural symmetries, could be imposed to them directly rather than as part of the gradient-based training loop. Secondly, real-space features of the

structural basis could be used at the analysis stage for automatic annotation of the latent space, e.g. through center of mass location or real-space PCA.

However, the use of the structural basis also implies a potential weakness in our method. If the heterogeneity of the data set cannot be embedded into a linear subspace of $K$ dimensions, our method would not be able to correctly reconstruct the volume manifold. This limit is theoretically not present in cryoDRGN, because they parameterize their reconstruction with a neural network. However, in practice, the capacity of the cryoDRGN network is also limited by computational cost, and we have so far not observed issues with the expressive capacity of our model. It should be noted that, due to the noise present in the cryo-EM reconstruction problem, one should expect that the dimension of the possible solution space $\hat{\mathcal{V}}$ is much smaller than $N$. Therefore, the limit to the reconstruction of heterogeneity due to noise would dominate the limit due to the representation capacity of the structural basis set.

# 7    Broader Impact

The cost of data processing can be a major limiting factor for research groups with limited access to computational resources. By improving robustness and computational efficiency, we expand the accessibility of the method to include a more diverse research community, which can in turn have an accelerating impact on the overall research quality of the field. Moreover, the potential for improved automation has implications for high-throughput structure determination. Along with improvements in microscope data acquisition, rapid processing can enable faster iterations in e.g. protein design or fragment-based drug discovery. Thereby, our method has the potential to impact medical research and the development of new therapeutics.

# 8    Acknowledgements

We are grateful to Johannes Schwab, Jasenko Zivanov and Daniel Mihaylov for helpful discussions, and to Jake Grimmett and Toby Darling for help with high-performance computing.

# 9    Funding

This work was funded by the UK Medical Research Council (MC_UP_A025_1013 to SHWS) and the European Union's Horizon 2020 research and innovation programme (under grant agreement no. 895412 to D.K.).

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
