## A    cryoDRGN-small Results

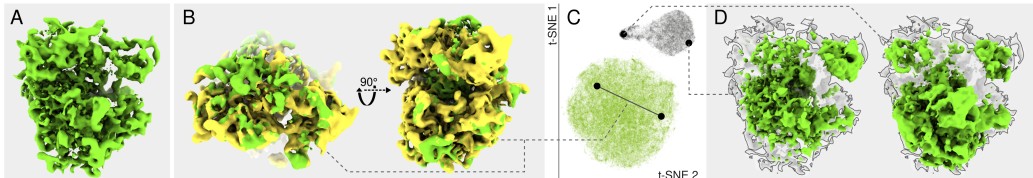

Figure 4: Reconstruction results of the ribosome data set using cryoDRGN-small settings. **(A)** Example reconstruction of the green region in the latent representation. **(B)** Comparison between two reconstructions with the largest numerical difference in real-space out of multiple samples randomly selected from the green cluster in the latent representation. No conformational variations, corresponding to known ribosomal states, could be observed. **(C)** The learnt latent representation. Cluster containing particle images of intact ribosomes marked in green and empty images or partially assembled ribosomes in grey. **(D)** Two example reconstruction results from the grey cluster.

## B    Extended FSC-diagram

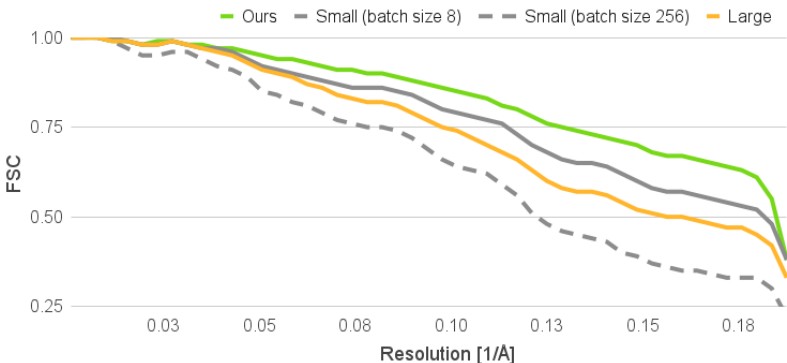

Figure 5: Extension of figure 2E, showing FSCs between final reconstruction results and ground truth for our method (green), cryoDRGN-small with batch size 8 (grey, solid) and batch size 256 (grey, dashed), and cryoDRGN-Large (orange).

## C    cryoDRGN Settings

### C.1    cryoDRGN Code

All cryoDRGN experiments mentioned in this paper were run with the following settings. The cryoDRGN program was cloned from the repository on Github, which is available at `https://github.com/zhonge/cryodrgn`. We used the commit with the hash `7c1e2d27b3ac76a5054cb6a662db8466910e9a51` belonging to branch `1.0.0-beta`, which was the latest commit at the time.

For experiments that mention *cryoDRGN-small*, the following input arguments were used:

```
--zdim 8 -n 50 --enc-dim 256 --enc-layers 3 --dec-dim 256 --dec-layers 3
```

For experiments with *cryoDRGN-large*, these arguments were used:

```
--zdim 8 -n 50 --enc-dim 1024 --enc-layers 3 --dec-dim 1024 --dec-layers 3
```

## C.2 cryoDRGN Parameters

For the Spike data set, cryoDRGN-small had 2,321,938 parameters in total: 2,048,016 for the encoder and 273,922 for the decoder. cryoDRGN-large has 14,006,290 parameters in total: 10,551,312 parameters for the encoder and 3,454,978 parameters for the decoder.

For the ribosome data set, cryoDRGN-small had 5,665,298 parameters in total: 5,342,224 for the encoder and 323,074 for the decoder. cryoDRGN-large has 27,379,730 parameters in total: 23,728,144 parameters for the encoder and 3,651,586 parameters for the decoder.

# D Runtime Statistics & Mean FSC

For the simulated data set, our method had a comparable memory footprint of 2.9 GiB compared to 2.01 GiB for cryoDRGN-small with batch size 8. However, running cryoDRGN with the same batch size as our method increased the memory footprint to 13.4 GiB.

For the experimental data set, our method had a smaller memory footprint of 4.5 GiB compared to 5.6 GiB for cryoDRGN-large with batch size 8. Running cryoDRGN with the same batch size as our method increased the memory footprint beyond the GPU capacity of 23.7 GiB. As an example, a batch size of 32 had a memory footprint of 16 GiB. See supplementary table 1 for more details.

Table 1: Runtime statistics for the simulated data set. cryoDRGN-large batch size 64 and Ours batch size 512, were aborted after one epoch only to measure memory and execution time. All times are in minutes. 'Time' refers to total run time and is only given for finished runs. 'Memory' refers to the peak measure GPU memory utilization, in GiB.

|  | Batch Size | Memory | Time | Time/epoch | Mean FSC |
|---|---|---|---|---|---|
| cryoDRGN-small | 8 | 2.01 | 213 | 4.2 | 0.46 |
| cryoDRGN-small | 16 | 2.42 | 126 | 2.5 | 0.44 |
| cryoDRGN-small | 256 | 13.4 | 48.7 | 0.97 | 0.39 |
| cryoDRGN-large | 8 | 3.0 | 414 | 8.3 | 0.43 |
| cryoDRGN-large | 64 | 11.4 |  | 5.7 |  |
| Ours | 256 | 2.9 | 40.3 | 0.81 | 0.49 |
| Ours | 512 | 3.3 |  | 0.73 |  |

Table 2: Runtime statistics for the Ribosome data set. All times are in minutes. 'Time' refers to total run time and is only given for finished runs. 'Memory' refers to the peak measure GPU memory utilization, in GiB.

|  | Batch Size | Memory | Time | Time/epoch |
|---|---|---|---|---|
| cryodrgn-small | 8 | 2.93 | 585 | 11.7 |
| cryodrgn-small | 64 | 11.9 |  | 5.9 |
| cryodrgn-large | 8 | 5.6 | 1638 | 32.8 |
| cryodrgn-large | 16 | 8.9 |  | 31.7 |
| cryodrgn-large | 32 | 16 |  | 30.6 |
| Ours | 256 | 4.5 | 210 | 4.2 |
| Ours | 512 | 5.2 |  | 3.9 |

# E Sparse Backpropagation Parameter Settings

For the simulated data set, the network used in our method had a parameter count of 13,820,388 for the encoder and 7,989,898 for the decoder, including the structural basis.

For the Ribosome data set, the network used in our method had a parameter count of 26,979,620 for the encoder and 36,783,274 for the decoder, including the structural basis.

In both runs the encoder is made up of five layers of a residual MLP and the decoder excluding the structural basis is a single linear layer that maps latent space dimensions to the number of structural basis, which is 16 for both cases.