# OpenReview forum: "Sparse Fourier Backpropagation in Cryo-EM Reconstruction"
_NeurIPS.cc/2022/Conference — NeurIPS 2022 Accept_

### Official Review · Reviewer_Vxds · 2022-07-11

**Rating:** 7
**Confidence:** 4
**Soundness:** 3 good
**Presentation:** 3 good
**Contribution:** 3 good

**Summary:**

The paper proposes an approach for learning a volumetric Fourier reconstruction in a VAE framework for heterogeneous reconstruction of SPA cryoEM. The authors bring attention to the fact that the previous related work of coordinate-based reconstructions can be improved by learning Fourier representations on a regular cubic grid. The new approach is more efficient and reconstructs with better FSC than current SOTA but at one-seventh of the computational cost.

**Questions:**

1. AFAIK, the paper or supplemental material makes no mention of the memory usage and the number of parameters. Moreover, you mention that the number of volumes for structural basis and volume size has major impact on memory footprint. Could you please expand on memory usage and at least approximately how it would scale?
2. Did you encounter any cases where your method fails or which you expect to fail? For example, do you believe the method would still work with low SNR datasets or datasets with many classes?

**Limitations:**

The authors adequately addressed the limitations and potential negative societal impact of their work.

**Strengths And Weaknesses:**

Strengths:
1. Faster and more efficient method makes cryoEM more accessible, allowing to iterate faster.
2. VAE framework, coupled with the previous point, allows interactive analysis of latent space (i.e. heterogeneouity), in contrast with discrete classification in the common cryoEM approaches.
3. The method is also better suited for cryoEM modality than previous coordinate-based networks.
4. The method is validated on both simulated and experimental datasets and shows better FSC than the closest related work.

Weaknesses:
1. No comparisons with current defacto standard cryoEM software packages like RELION and cryoSPARC. For practical usage in cryoEM, it would be necessary to compare the proposed method with current standards.
2. The method expects projection angle, shift and CTF parameters to be provided. The authors however mention that it should be doable to estimate them and leave it for the future work.

---

> ### Author Response · Authors · 2022-08-02
> **More statistics on memory and models added, mentions on problematic cases**
>
> No comparisons with current de facto standard cryoEM software packages like RELION and cryoSPARC. For practical usage in cryoEM, it would be necessary to compare the proposed method with current standards.
>
> **We used RELION 3D classification to find the ‘garbage’ particles in the Ribosome data set. This repeated the same calculations performed in the original paper for this data set (see Wilson, et al., eLife 2014). Generally, it is difficult to assess 3D classification results of the intact particles when the variability is continuous like what is seen in the Ribosome data set. We will clarify this in the text.**
>
> ***
>
> The method expects projection angle, shift and CTF parameters to be provided. The authors however mention that it should be doable to estimate them and leave it for the future work.
>
> **One of the benefits of the presented method is the fast full volume inference. This can be used for global angular searches, which is an important part of standard software reconstruction algorithms. To do this efficiently, we need to implement GPU kernels that can be integrated into a python/pytorch context.**
>
> **However, our experiments suggest that simultaneous local refinement of poses and CTF parameters based on gradient backpropagation cause more overfitting to noise, which had detrimental effects on the reconstruction. Hence we need to implement better regularization, which goes beyond the scope of this work.**
>
> ***
>
> AFAIK, the paper or supplemental material makes no mention of the memory usage and the number of parameters. Moreover, you mention that the number of volumes for structural basis and volume size has major impact on memory footprint. Could you please expand on memory usage and at least approximately how it would scale?
>
> **We have now added the memory footprint in the supplementary section E and the parameter count of our method in supplementary section G. For the cryoDRGN parameter count, please refer to supplementary section C.2.**
>
> ***
>
> Did you encounter any cases where your method fails or which you expect to fail? For example, do you believe the method would still work with low SNR datasets or datasets with many classes?
>
> **At lower SNRs overfitting becomes more of a problem. In the latest version of the manuscript we have reduced the SNR of the simulated data set to make it more similar to experimental data. CryoDRGN shows signs of overfitting when the decoder capacity is increased. Our method seems to be less susceptible to noise in these experiments, but at some point our method will also overfit. Hence, work needs to be done on regularization similar to what has been done in standard cryoEM software to prevent this. If the reviewer wishes, we can add this to the discussion section.**

---

> > ### Comment · Reviewer_Vxds · 2022-08-09
> > **Thanks!**
> >
> > Thank you for addressing reviewers’ questions and well done additions to the manuscript. Now even more I stand by my initial rating and believe this paper is a valuable contribution to the community.

---

### Official Review · Reviewer_WYec · 2022-07-11

**Rating:** 7
**Confidence:** 3
**Soundness:** 3 good
**Presentation:** 3 good
**Contribution:** 3 good

**Summary:**

This paper proposes to replace the coordinate-based Fourier space cryo-EM reconstruction approach in CryoDRGN with a volumetric Fourier space representation. The idea is that this volumetric representation allows the model to capture high-frequency variation typical of Fourier space data better than coordinate-based networks, which implicitly tend to captures smoother structures. While the volumetric representation increases memory requirements especially during backpropagation, the paper introduces a sparse back propagation strategy which takes advantage of the sparsity of the projection operation in the cryo-EM forward model. The paper demonstrates similar or better performance on reconstructing two datasets of cryo-EM images while requiring significantly shorter training and inference times, because forward passes do not require individual inference runs for each coordinate.

**Questions:**

- What aspects of the visual results lead to the claim that the reconstruction quality is improved with the proposed method (lines 256-257)?
- How is the tri-linear interpolation able to handle the very high frequency changes in Fourier-space data which is ostensibly problematic for the coordinate-based approach?

[My suggestion may change based on the responses to the above two questions, but as it currently stands, my suggestion is to remove claims about improved performance while focusing on the computational savings, which are more thoroughly justified. Or, if results using spatial priors capitalizing on the volumetric representation significantly improve the results, that could strengthen these claims -- but I don't think these experiments are necessary for publication of the paper as is.]

**Limitations:**

The authors describe a key limitation of their method in the last paragraph of Section 6 and address societal impacts in the "Broader Impacts" section. I believe that the proposed method has higher memory requirements than cryoDRGN, which I would encourage the authors to discuss more explicitly in the Discussion section as well.

**Strengths And Weaknesses:**

# Strengths
- The paper is quite clearly written and provides enough background for a reader outside of cryo-EM to easily understand the problem setup and categories of competing approaches, as well as some of their tradeoffs.
- The reported savings in computational resources from the new method are significant, intuitively make sense (in that it is no longer necessary to run individual inference passes for each coordinate), and would be very useful for users of cryo-EM reconstruction tools.
- The sparse backpropagation scheme is clever and makes use of the volumetric representation effectively.

# Weaknesses
- **It is not clear that the reconstruction quality of the proposed method is significantly higher than that of cryoDRGN.** This may be because I am not an expert in reading these kinds of images, and I am willing to change my mind on this. But as the manuscript currently stands, I can see differences between Figure 2B and 2D, but I’m not sure how to tell which is correct or why the proposed method reconstructions are better. Perhaps it would be helpful to point these out explicitly with arrows or more detailed text descriptions. Similarly, I am not sure what the salient takeaways are from the visual results in Figure 3. The FSC plot in 2E does show slightly higher FSC correlation than the cryoDRGN method, but this difference appears to decrease at high resolution, where I presume these methods would be most useful.
- **It is not clear that the proposed method makes it easier for the networks to capture “local, high-frequency changes.”** While I agree that the coordinate-based approaches may suffer from this issue, it is not clear to me that the proposed approach is significantly better — in particular, I imagine the tri-linear interpolation (lines 204-205) might correspondingly struggle to handle very high frequency changes in the Fourier-space data. Perhaps the tri-linear interpolation does provide good results if the high-frequency changes are greater than ~1 pixel in width, but if so, I would expect the coordinate methods to also capture this reasonably well. This is further supported by my previous point that it is not clear that the proposed method results in significantly higher reconstruction quality, so at a minimum, some more discussion of this issue is needed.
- (Minor) It may be helpful to report the real memory usage of both CryoDRGN and the proposed method so that readers can understand this tradeoff a bit more concretely. Relatedly, I see the parameter counts for cryoDRGN in Appendix C.2, but do not see these for the proposed method -- please provide these to compare.  This will give a rough sense for the expressive capacity of the models relative to each other (and, depending on the numbers, may make the point about capacity in lines 339-348 more concrete).

---

> ### Author Response · Authors · 2022-08-02
> **Improved figures and text, more details on correlations in DTFT**
>
> It is not clear that the reconstruction quality of the proposed method is significantly higher than that of cryoDRGN. This may be because I am not an expert in reading these kinds of images, and I am willing to change my mind on this. But as the manuscript currently stands, I can see differences between Figure 2B and 2D, but I’m not sure how to tell which is correct or why the proposed method reconstructions are better. Perhaps it would be helpful to point these out explicitly with arrows or more detailed text descriptions. [...] The FSC plot in 2E does show slightly higher FSC correlation than the cryoDRGN method, but this difference appears to decrease at high resolution, where I presume these methods would be most useful.
>
> **At high frequencies (higher resolution than 5.68) the FSC drops for both methods. This is due to aliasing artifacts close to the Nyquist limit. We have clarified this both in the text and the figure. We now plot the calculated maximum possible FSC with ground truth poses and conformational assignments. One can clearly see that also this FSC drops at the Nyquist frequency. Additionally, we have clarified where to look in the figure to best see differences in the map quality by adding the contour of the ground truth for each conformation. Discrepancies are most apparent in the flexible regions in the cryoDRGN results, which is typically the most important part of reconstruction when applying this type of methods. Since these regions make up a smaller portion of the volume their effect on the global FSC is smaller. In hindsight, a simulated data set with a larger moving part would have been better suited to highlight the improvement in the FSC.**
>
> ***
>
> It is not clear that the proposed method makes it easier for the networks to capture “local, high-frequency changes.” While I agree that the coordinate-based approaches may suffer from this issue, it is not clear to me that the proposed approach is significantly better — in particular, I imagine the tri-linear interpolation (lines 204-205) might correspondingly struggle to handle very high frequency changes in the Fourier-space data. Perhaps the tri-linear interpolation does provide good results if the high-frequency changes are greater than ~1 pixel in width, but if so, I would expect the coordinate methods to also capture this reasonably well. [...]
>
> **Increasing the sampling in a DTFT is equivalent to padding the real space volume with zeros (please see the zero padding theorem). One could imagine using cryoDRGN to reconstruct volumes with higher Fourier space sampling and then cropping in real space to get back to the original volume. Doing so could potentially improve the results in certain data sets with high defocus CTFs. But this can also be accomplished in our method by increasing the box size through zero padding, although this would require a larger increase in computational cost. However, this would only make a difference at reconstruction resolutions beyond what is typically needed for heterogeneity analysis.**
>
> ***
>
> It may be helpful to report the real memory usage of both CryoDRGN and the proposed method so that readers can understand this tradeoff a bit more concretely. Relatedly, I see the parameter counts for cryoDRGN in Appendix C.2, but do not see these for the proposed method [...] This will give a rough sense for the expressive capacity of the models relative to each other (and, depending on the numbers, may make the point about capacity in lines 339-348 more concrete).
>
> **We have appended the measured memory usage of each method to supplementary section E. Also, the parameter count for our method is appended in supplementary section G.**
>
> ***
>
> What aspects of the visual results lead to the claim that the reconstruction quality is improved with the proposed method (lines 256-257)?
>
> **This should now be easier to see in figure 2. Please also see the above comments.**
>
> ***
>
> How is the tri-linear interpolation able to handle the very high frequency changes in Fourier-space data which is ostensibly problematic for the coordinate-based approach?
>
> **Neighboring coefficients in a DTFT with optimal sampling are generally not correlated. This results in high frequency variations in the output space of the decoder, which has to be handled with a high frequency positional embedding in cryoDRGN. In our approach, the sampling of Fourier space is optimal based on DTFT theory. Increasing sampling will only introduce redundancy in the form of correlated coefficients. This supports the use of trilinear interpolation for any points between the sampling points, since they are highly correlated with the nearby sampling points if these are optimally spaced.**
>
> **Please also see the fourth item in [this comment](https://openreview.net/forum?id=51f5sPXJD_E&noteId=p3h9TPKrYZd) for more details.**

---

> > ### Comment · Reviewer_WYec · 2022-08-08
> > **Thanks for the response; just 1 minor question/clarification remaining.**
> >
> > Thanks to the authors for their detailed, point-by point responses and edits to the manuscript. The changes to Figure 2, especially, make it much easier to understand qualitative improvements from this method.
> >
> > I just have one outstanding clarification. As mentioned in this response, neighboring coefficients in a DTFT with optimal sampling are generally not correlated. Why is trilinear interpolation, and not sinc interpolation, the appropriate thing to do here? Perhaps this is an ease of implementation issue -- in any case, I don't think this issue is critical for acceptance, as the trilinear approximation seems to give sufficiently good results.
> >
> > I have also read the other reviews and am tentatively satisfied with the responses, especially with the additional architectural details and modified simulation (though I'm not very experienced with this type data and would like to hear that reviewer's response on whether the new simulations are acceptable). Overall, I will keep my rating at "Accept."

---

> > > ### Author Response · Authors · 2022-08-09
> > > **Sinc interpolation**
> > >
> > > We thank the reviewer for their suggestions, which have improved the manuscript.
> > >
> > > The reviewer is correct in that sinc interpolation is theoretically the most accurate interpolation, since it is equivalent to zero-padding (please see [this]( https://ccrma.stanford.edu/~jos/resample/Theory_Ideal_Bandlimited_Interpolation.html)). The primary reason to choose trilinear interpolation is computational cost. Linear interpolation is extremely local in its memory access and requires fewer operations compared to sinc interpolation. Additionally, since the backward pass would become complicated for sinc interpolation, there’s ease of implementation, as mentioned by the reviewer. However, since sinc interpolation is equivalent to zero padding, one could improve the approximation of linear interpolation by zero padding all the transforms. At sufficient zero padding, any numerical differences between sinc and linear interpolation should become negligible. This is a point we will add to the camera ready version.

---

> > > > ### Comment · Reviewer_WYec · 2022-08-09
> > > > **Thanks!**
> > > >
> > > > Thanks for this additional clarification, and for adding a note about this to the manuscript.

---

### Official Review · Reviewer_4A7D · 2022-07-16

**Rating:** 4
**Confidence:** 5
**Soundness:** 2 fair
**Presentation:** 3 good
**Contribution:** 2 fair

**Summary:**

This paper presents a method for heterogeneous cryo-EM reconstruction that fits 3D density volumes as a linear combination of voxel arrays where scale factors are inferred with an encoder network in a VAE framework. The model for volumes is represented in Fourier space and trained on projection images via the Fourier slice theorem (i.e. as 2D planar slices of 3D volumes) to reduce the memory requirement for backpropagation. The authors compare their method against cryoDRGN (a VAE-based model with a coordinate MLP decoder) and show slightly higher resolution reconstructions with their voxel-based generative model on a small synthetic dataset. The method yields qualitatively similar results to cryoDRGN on an experimental dataset of the Plasmodium falciparum 80S ribosome.

**Questions:**

* How does this method compare to 3DVA on the tested datasets? Can the authors discuss the advantages/disadvantages of this method compared to 3DVA?
* How is the model initialized and how much does this impact the convergence/accuracy of the reconstruction?
* On the synthetic dataset, the reconstruction accuracy of just a single volume is quantified. It would be good to verify that the accuracy is consistent across all the different states.
* There are some architectural and training details that are missing, e.g. the size of the network for encoding images and decoding scale factors. Training details like the batch sizes are also missing, which has a huge impact on training times, especially when comparing methods by the number of epochs trained.


**Limitations:**

Yes

**Strengths And Weaknesses:**

Strengths:
* Novel formulation of cryo-EM heterogeneous reconstruction, which mixes and matches the generative model from 3DVA (linear combination of voxel arrays) and inference method from cryoDRGN (variational encoder MLP).
* Rendering volumes is fast and interpretable, i.e. a linear combination of K 3D voxel arrays.
* The paper is well written and easy to follow

Weaknesses:

While the method is interesting and could become a useful tool, the evaluation is flawed/limited.

* The authors state that one of the main advantages is the computational cost for training (218 min for their method vs. 1638 min for cryoDRGN), but the training times are reported for a fixed number of training epochs without considering the convergence rate of the different methods.
* Testing on synthetic data is useful for validating the method since there is ground truth, but the synthetic dataset used here seems a little too simple to be realistic (only 10k images of 10 discrete states at a very high signal-to-noise ratio of 2/3).
* The authors mention a novel backpropagation method, but there is very little detail on what this method is. Is this just using the Fourier slice theorem to render 2D slices instead of 3D volumes (e.g. in cryoDRGN and in the voxel-based method described in Ullrich et all UAI 2019). Could the authors elaborate?

A minor comment:
* I don't follow some aspects of the motivation (Section 4.1). The authors point out that the cryo-EM signal is high-frequency in the Fourier domain and coordinate MLPs like cryoDRGN/NERF poorly capture high frequency signals unless a high-frequency positional encoding is provided in the input. But cryoDRGN already uses a positional encoding for this very reason, so what's the issue? Later in the results section, much of the results are instead focused on the training/render time of a voxel-based representation. The authors also discuss at length many advantages of a real space formulation, which further confuses the motivation of the work (a Fourier space representation).

---

> ### Author Response · Authors · 2022-08-02
> **New, more realistic simulated dataset and extended experiments**
>
> The authors state that one of the main advantages is the computational cost for training [...]
>
> **We have added supplementary figure 6 that plots the MSE loss for each run, which shows that neither method reaches convergence within the 50 epochs for either dataset.**
>
> ***
>
> Testing on synthetic data is useful for validating [...]
>
> **We have now replaced the simulated dataset with one that contains images with a more realistic SNR (1/10) and number of particles (100k). This should make the overall statistics of the synthetic dataset more similar to that of experimental data. Please see updates to section 5.1 for more details on this.**
>
> ***
>
> The authors mention a novel backpropagation method, but there is very little detail on what this method is [...]
>
> **The Ullrich et al. paper describes the trilinear interpolation and its differentiation well. We will add this reference and further clarified the details.**
>
> ***
>
> I don't follow some aspects of the motivation (Section 4.1) [...]
>
> **We believe it is illustrative to think of our approach as an “explicit coordinate” approach, as opposed to the “implicit coordinate” approach of NeRF based decoders like cryoDRGN. This means that for some coordinate $(x,y,z)$, cryoDRGN represents this in the weights of its decoder and will have to compute the value of the volume for some latent space encoding $l$ with the result of evaluating the neural network $f(l,x,y,z)$. This is natural in real space rendering, as we expect that whenever we have points close to $(x,y,z)$ in the output space, the value of the output would also be close to$ f(l,x,y,z)$. That is, we expect the solution to be a Lipschitz continuous function with a low Lipschitz constant. However, the output of this function is in Fourier space. In general, there is no correlation between neighboring Fourier components in a DTFT, so even for another coordinate $(x’,y’,z’)$ that is very close to $(x,y,z)$, it might be the case that the difference $||f(l,x,y,z) - f(l,x’,y’,z’)||$ is large. There has been work to show that high Lipschitz constants have adverse effects (or are indicative of adverse effects) on generalization (this paper includes some background https://proceedings.neurips.cc/paper/2019/file/95e1533eb1b20a97777749fb94fdb944-Paper.pdf). This issue can be somewhat ameliorated by adding a high frequency positional encoding so that even if $(x,y,z)$ and $(x’,y’,z’)$ are close, $p(x,y,z)$ and $p(x’,y’,z’)$ are far. But this only makes the denominator of the Lipschitz constant calculation larger, it is not clear what negative impacts this may have on convergence. We believe our explicit coordinate approach, where the volumes are represented on a grid to be more powerful. This allows the close-by indices in the output to be fully uncorrelated, side-stepping this issue completely.**
>
> **We feel this detailed description goes somewhat beyond the scope of the manuscript. But if the reviewer insists, we could add it to the camera-ready version of this paper.**
>
> ***
>
> Later in the results section, much of the results are instead focused on [...]
>
> **The proposed method enables faster full volume inference and gradient backpropagation, which makes it possible to efficiently impose real space 3D priors. As an example, solvent masks require an inverse 3D Fourier transform to be applied. This could be done either on the structural bases directly or with an auxiliary loss as part of the gradient. We can clarify this in the text.**
>
> ***
>
> How does this method compare to 3DVA [...]
>
> **Because the presented method has more similarities to cryoDRGN than 3DVA, we decided to limit comparison of the results to cryoDRGN only. We can amend section 4 to further elaborate on the theoretical differences to 3DVA.**
>
> ***
>
> How is the model initialized [...]
>
> **The encoder and decoder are initialized with default pytorch initialization schemes. The structural bases are initialized with random values from a normal distribution for low frequency components and zeros for high frequency components. The structural basis bias is initialized to zeros. Also initializing the high frequencies with random values from a normal distribution had detrimental effects on convergence, when applied to the simulated data set. We will add this clarification to section 4 in the camera-ready version.**
>
> ***
>
> On the synthetic dataset, the reconstruction accuracy [...]
>
> **With the new simulated data set we now evaluate the FSC on samples from all configurations across all three states. The reported FSC is an average over these. Please see figure 2.**
>
> ***
>
> There are some architectural and training details that are missing [...]
>
> **We have added more details about this in the supplementary. Please see supplementary table 1 and 2 in sections E and description in section G. As well as results presented in section 5.1 and 5.2.**

---

> > ### Comment · Reviewer_4A7D · 2022-08-09
> > **Improved synthetic experiments but flawed measure of convergence and advantage over 3DVA?**
> >
> > Thank you for the clarifications, additional details and for repeating the synthetic data experiments with a more realistic noise level and dataset size. Overall I think this method could be a valuable contribution to the community, but I do think there are still some flaws in the evaluation of convergence and a lack of discussion/comparison to 3D variability analysis (3DVA) [1]. Please consider my comments below.
> >
> > [convergence metric / timing comparisons]
> > Reporting the loss on the training dataset will not reflect convergence of the model (e.g. it can always decrease by overfitting to noise). Since these experiments are performed on synthetic data, I think the most appropriate metric is FSC to the ground truth volume at each epoch. It may be that this voxel based approach will converge earlier than the neural field representation.
> >
> > [comparison to 3dva]
> > The method is more similar to 3DVA than cryoDRGN in terms of the generative model (linear combination of voxel arrays) and thus the expressive capabilities for heterogeneity. Especially since 3DVA is currently more popular than cryoDRGN among cryo-EM users, I think it is important to at least include a discussion of the differences between this method and 3DVA. What is the advantage of this approach relative to 3DVA? Ideally, a benchmark against 3DVA would make the paper a stronger contribution to the community.
> >
> > [rational for neural field vs. voxel based model]
> > Thanks for the clarification. Including the full explanation is not necessary, though it may be nice to mention the limitations of this voxel-based generative model (inaccurate approximation of nonlinear motions).
> >
> > [another suggestion for improvement]
> > It would be more convincing to include results on a second experimental dataset. Consider the pre-catalytic spliceosome dataset (EMPIAR-10180) with continuous heterogeneity that has been benchmarked in several methods papers.
> >
> > [1] Punjani & Fleet. 3D variability analysis: Resolving continuous flexibility and discrete heterogeneity from single particle cryo-EM. JSB 2021

---

> > > ### Author Response · Authors · 2022-08-09
> > > **Good points for further improvements**
> > >
> > > We thank the reviewer for the detailed comments and the follow-up discussion.
> > >
> > > [convergence metric / timing comparisons] This is a good point. We will also include the FSC to the ground truth at each epoch in the camera-ready version of the manuscript. Once we have these numbers, we will also submit them as a comment to this OpenReview submission.
> > >
> > > [comparison to 3dva] We are in the process of writing a more expansive comparison to 3DVA, which will be included in the camera-ready version of the manuscript. The main advantage of our method is that the mixing coefficients of the linear model are provided by a neural network and so it may be able to better capture heterogeneity than combinations of eigenvectors. Time permitting, we can also run a comparison to 3DVA and include it in the camera-ready manuscript.
> > >
> > > [spliceosome] We believe that one simulated dataset and one experimental dataset should be sufficient to showcase the differences between the methods. However, if the reviewer believes this to be important to our case, then we can also run this dataset.

---

### Author Response · Authors · 2022-08-09
**FSC Convergence Plot**

This is the FSC data, ordered by epoch in response to reviewer 4A7D. The plotted version of this data will be included in the camera-ready version of the manuscript. The difference in convergence is quite clear in these graphs as well.
In order to obtain this data, three particles of each state were reconstructed using cryoDRGN large, small, and sparse backprop. The FSC was calculated to the ground truth states and then averaged.
There are two versions of this data: one is based on the resolution at which the FSC falls below 0.7 and the second is the average FSC across all shells .

Resolution at 0.7 Threshold
==================================================
| epoch | cryoDRGN large | cryoDRGN small | sparse backprop |
|-------|----------------|----------------|-----------------|
| 1     | 9.846277       | 11.130573      | 9.142971        |
| 2     | 8.827696       | 10.6668        | 7.757672        |
| 3     | 8.53344        | 10.240128      | 8.0001          |
| 4     | 8.53344        | 10.240128      | 8.0001          |
| 5     | 8.258167       | 9.846277       | 8.0001          |
| 6     | 8.258167       | 9.846277       | 8.0001          |
| 7     | 8.258167       | 9.846277       | 8.0001          |
| 8     | 8.258167       | 9.142971       | 8.0001          |
| 9     | 8.258167       | 9.142971       | 8.0001          |
| 10    | 8.258167       | 8.827696       | 7.757672        |
| 11    | 8.258167       | 8.827696       | 7.757672        |
| 12    | 8.258167       | 8.827696       | 7.757672        |
| 13    | 8.258167       | 8.53344        | 7.757672        |
| 14    | 8.258167       | 8.53344        | 7.314377        |
| 15    | 8.258167       | 8.53344        | 7.757672        |
| 16    | 8.258167       | 8.53344        | 7.529506        |
| 17    | 8.258167       | 8.53344        | 6.919005        |
| 18    | 8.258167       | 8.53344        | 7.529506        |
| 19    | 8.258167       | 8.53344        | 7.1112          |
| 20    | 8.258167       | 8.53344        | 7.1112          |
| 21    | 8.258167       | 8.53344        | 7.529506        |
| 22    | 8.53344        | 8.258167       | 6.919005        |
| 23    | 8.53344        | 8.258167       | 7.1112          |
| 24    | 8.53344        | 8.258167       | 6.919005        |
| 25    | 8.53344        | 8.258167       | 7.314377        |
| 26    | 8.53344        | 8.258167       | 6.919005        |
| 27    | 8.53344        | 8.258167       | 6.919005        |
| 28    | 8.53344        | 8.258167       | 7.1112          |
| 29    | 8.53344        | 8.258167       | 7.1112          |
| 30    | 8.53344        | 8.258167       | 6.919005        |
| 31    | 8.53344        | 8.258167       | 6.919005        |
| 32    | 8.53344        | 8.258167       | 6.919005        |
| 33    | 8.53344        | 8.258167       | 6.919005        |
| 34    | 8.53344        | 8.258167       | 6.919005        |
| 35    | 8.827696       | 8.258167       | 7.1112          |
| 36    | 8.827696       | 8.258167       | 6.919005        |
| 37    | 8.53344        | 8.258167       | 6.919005        |
| 38    | 8.827696       | 8.258167       | 6.736926        |
| 39    | 8.827696       | 8.258167       | 6.919005        |
| 40    | 8.827696       | 8.258167       | 7.1112          |
| 41    | 8.827696       | 8.258167       | 6.919005        |
| 42    | 8.827696       | 8.258167       | 6.919005        |
| 43    | 8.827696       | 8.258167       | 6.919005        |
| 44    | 8.827696       | 8.258167       | 6.736926        |
| 45    | 8.827696       | 8.258167       | 6.736926        |
| 46    | 8.827696       | 8.0001         | 6.736926        |
| 47    | 8.827696       | 8.258167       | 6.736926        |
| 48    | 8.827696       | 8.0001         | 6.736926        |
| 49    | 9.142971       | 8.0001         | 6.736926        |

---

> ### Author Response · Authors · 2022-08-09
> **Continued**
>
> Average FSC
> ==============================
> | epoch | cryoDRGN large | cryoDRGN small | sparse backprop |
> |-------|----------------|----------------|-----------------|
> | 1     | 0.4026649      | 0.36271226     | 0.41322434      |
> | 2     | 0.42484552     | 0.37943053     | 0.4662058       |
> | 3     | 0.4324309      | 0.390301       | 0.45662948      |
> | 4     | 0.4456272      | 0.39700902     | 0.45653507      |
> | 5     | 0.44882172     | 0.40347958     | 0.46001554      |
> | 6     | 0.44950223     | 0.40448815     | 0.46212682      |
> | 7     | 0.45123887     | 0.40837213     | 0.4627513       |
> | 8     | 0.45443353     | 0.41553697     | 0.45934224      |
> | 9     | 0.45549354     | 0.41696817     | 0.4623152       |
> | 10    | 0.45343038     | 0.42067996     | 0.4638603       |
> | 11    | 0.4546782      | 0.42341504     | 0.46770892      |
> | 12    | 0.45284423     | 0.42380884     | 0.46691453      |
> | 13    | 0.45406988     | 0.42857307     | 0.46388647      |
> | 14    | 0.4514567      | 0.42937428     | 0.47306585      |
> | 15    | 0.4518208      | 0.4307488      | 0.46774215      |
> | 16    | 0.4523333      | 0.43113413     | 0.47200483      |
> | 17    | 0.45020616     | 0.43362355     | 0.4787152       |
> | 18    | 0.4489091      | 0.43657827     | 0.4716727       |
> | 19    | 0.44888878     | 0.43581125     | 0.47516575      |
> | 20    | 0.4484283      | 0.43700188     | 0.47550106      |
> | 21    | 0.44968262     | 0.4391308      | 0.47118685      |
> | 22    | 0.44754586     | 0.44167903     | 0.48022935      |
> | 23    | 0.44585463     | 0.44362056     | 0.4752361       |
> | 24    | 0.44702634     | 0.44623998     | 0.48185235      |
> | 25    | 0.44424608     | 0.44344518     | 0.4728565       |
> | 26    | 0.44542778     | 0.44413495     | 0.47786087      |
> | 27    | 0.44479364     | 0.4446329      | 0.48055094      |
> | 28    | 0.44344908     | 0.4461772      | 0.4761763       |
> | 29    | 0.44261727     | 0.44845134     | 0.47736934      |
> | 30    | 0.44274417     | 0.44966784     | 0.48052537      |
> | 31    | 0.4416719      | 0.44957528     | 0.481153        |
> | 32    | 0.44045183     | 0.45111632     | 0.4810452       |
> | 33    | 0.44095886     | 0.45107546     | 0.4799954       |
> | 34    | 0.44242516     | 0.45161793     | 0.48105642      |
> | 35    | 0.44010815     | 0.45461136     | 0.47639814      |
> | 36    | 0.43911287     | 0.45338494     | 0.48056117      |
> | 37    | 0.43909216     | 0.45147055     | 0.48290464      |
> | 38    | 0.43840322     | 0.4533277      | 0.48406294      |
> | 39    | 0.43826973     | 0.4538183      | 0.4796279       |
> | 40    | 0.43777597     | 0.45664445     | 0.47784445      |
> | 41    | 0.43751425     | 0.45644695     | 0.48049268      |
> | 42    | 0.43784016     | 0.45592365     | 0.48146236      |
> | 43    | 0.4355459      | 0.45566455     | 0.48335156      |
> | 44    | 0.43651858     | 0.45500138     | 0.48510268      |
> | 45    | 0.43603167     | 0.45656747     | 0.48365855      |
> | 46    | 0.4348933      | 0.4573528      | 0.4856207       |
> | 47    | 0.43528965     | 0.4543395      | 0.4845828       |
> | 48    | 0.43400434     | 0.45868832     | 0.48733646      |
> | 49    | 0.43371433     | 0.4585916      | 0.4866522       |

---

### Meta-Review · Area_Chair_myqG · 2022-09-10

**Recommendation:** Accept
**Confidence:** Certain

**Metareview:**

The paper presents a new cryoEM reconstruction algorithm for data with multiple structural states for the same protein. In contrast to previous approaches which use a coordinate-based implicit representation of reconstructed density, this paper reconstructs an explicit volumetric grid and the optimization is formulated in the Fourier domain. Good accuracy and faster convergence is demonstrated, in comparison to prior methods.

While all reviewers agreed that the method constituted an important contribution to the field, they also pointed to shortcomings in the evaluation of the method. In response, the authors added analysis of the convergence based on FSC-based resolution estimates, and offered to add a comparison to the 3DVA which uses a similar representation to the proposed method.

Based on the reviews and author responses, I recommend acceptance of this paper. I urge the authors to follow through on "writing a more expansive comparison to 3DVA, which will be included in the camera-ready version of the manuscript".

**Award:**

No

---

### Decision · Program_Chairs · 2022-09-14

Accept